# *Leathesia difformis* Extract Inhibits α-MSH-Induced Melanogenesis in B16F10 Cells via Down-Regulation of CREB Signaling Pathway

**DOI:** 10.3390/ijms20030536

**Published:** 2019-01-28

**Authors:** Ga-Young Seo, Yuna Ha, Ah-Hyun Park, Oh Wook Kwon, Youn-Jung Kim

**Affiliations:** 1Research Institute of Basic Sciences, Incheon National University, Incheon 22012, Korea; tjrkdud7011@naver.com (G.-Y.S.); dbsk335@daum.net (Y.H.); dkgus3021@naver.com (A.-H.P); hades770@hanmail.net (O.W.K.); 2Department of Cosmetic Science and Management, Graduate school, Incheon National University, Incheon 22012, Korea; 3Department of Marine Sciences, Incheon National University, Incheon 22012, Korea

**Keywords:** *Leathesia difformis*, melanogenesis, tyrosinase, CREB pathway, B16F10

## Abstract

*Leathesia difformis* (L.) Areschoug (*L. difformis*) is a species of littoral brown algae of the class Phaeophyceae. Only a few studies on the apoptotic, antiviral, and antioxidant properties of *L. difformis* have been reported, and its inhibitory effect on melanin synthesis has not been studied. The aim of this study was to investigate the anti-melanogenic effect of *L. difformis* extract on α-melanocyte-stimulating hormone (α-MSH)-induced B16F10 melanocytes and its mechanism of action. *L. difformis* was extracted using 80% ethanol (LDE) and then fractioned between ethyl acetate (LDE-EA) and water (LDE-A). Our data demonstrated that LDE-EA significantly inhibited melanin level and cellular tyrosinase activity in α-MSH-stimulated B16 cells. In addition, the expression of genes associated with melanin synthesis, such as microphthalmia-associated transcription factor (*Mitf*), tyrosinase (*Tyr*), tyrosinase-related protein-1 (*Trp-1*), dopachrome tautomerase (*Dct*), and melanocortin 1 receptor (*Mc1r*) was down-regulated by LDE-EA treatment. Moreover, LDE-EA decreased p-CREB levels, which suggests that the inhibition of the cAMP/PKA/CREB pathways may be involved in the anti-melanogenic effect of LDE-EA. Thus, this study revealed that LDE-EA is an effective inhibitor of hyperpigmentation through inhibition of CREB pathways and may be considered as a potential therapeutic agent for hyperpigmentation disorders.

## 1. Introduction

Melanocytes are located in the stratum basale of the epidermis. Melanocytes produce melanin in the melanosomes and transfer this to the keratinocytes, which gradually migrate to the outer layer of the skin. Human skin color is determined by the melanin content in the transferred keratinocytes [1,2]. In addition, melanin biosynthesis is crucial to enhance the body’s defense against harmful effects of ultraviolet (UV) radiation and DNA damage to the skin [3,4]. However, excessive melanin production can lead to hyperpigmentation in the form of spots, freckles, and inflammation due to prolonged exposure to UV light [5,6].

The regulation of melanogenesis is by growth factors, hormones, cytokines, and UV. When the skin (melanocytes) is exposed to UV light, keratinocytes produce adrenocorticotropic hormone, α-melanocyte stimulating hormone (α-MSH), and endothelin-1, which indirectly increases the production of melanin [4,7], and α-MSH subsequently binds to melanocortin-1 receptor (Mc1r), a G-protein-coupled membrane receptor expressed only in melanocytes and secreted from the middle cerebral pituitary gland to activate adenylate cyclase. This process amplifies intracellular cyclic adenosine monophosphate (cAMP) signaling and activates protein kinase A (PKA) and intracellular cAMP-response element binding (CREB) protein [8,9,10]. CREB activated through phosphorylation increases the expression of microphthalmia-associated transcription factor (MITF), which is specific to melanocyte transcription [11,12].

Melanin synthesis is regulated by enzymes such as tyrosinase, tyrosinase-related protein 1 (TRP-1), and dopachrome tautomerase (DCT). Tyrosinase is an enzyme that regulates the rate-limiting step of melanin production, which oxidizes 3,4-dihydroxy phenylalanine (DOPA) into dopaquinone. TRP-1 oxidizes 5,6-dihydroxyindole-2-carboxylic acid (DHICA) to red indole-5,6-quinone-2-carboxylic acid and regulates tyrosinase activity. DCT converts DOPA chrome to DHICA, which is subsequently converted into eumelanin and pheomelanin [13,14,15]. Therefore, in vitro assays that can efficiently measure the inhibition of activities of MITF, tyrosinase, TRP-1, and DCT are crucial for the evaluation of skin-whitening preparations [16].

The most commonly used whitening agents include hydroquinone, arbutin, kojic acid, ascorbic acid, and retinoic acid, but there are some reports that they can have toxic and allergic effects on the skin [17,18]. For example, hydroquinone inhibits tyrosinase activity in melanin formation and improves pigmentation and formation by interfering with the oxidation process of tyrosine in the early stages of melanin biosynthesis. However, it can cause skin irritation and contact dermatitis, and it has been reported that various side effects, such as ochronosis, are observed when used for a long period of time, and it is prohibited to use as a raw material for whitening products, and only a part thereof is used as a medicine by prescription [19,20,21]. In addition, kojic acid and ascorbic acid inhibit tyrosinase activity in melanin synthesis and inhibit the final stage of melanin production. However, it is not only low in skin penetration, stability, and whitening effect, but it can cause cytotoxicity, dermatitis, and erythema in long term use [22,23]. In this regard, there is an increasing need to develop safe and effective whitening agents with minimal side effects.

Seaweeds exert few toxic effects and contain various physiologically active substances; thus, they are widely used as medicines, health foods, and functional cosmetics. In particular, brown alga contains copious quantities of laminaran and fucoidan, polysaccharides known to exert anticancer, antioxidant, and blood coagulation effects. Studies on the skin-related activities of marine algae have been conducted in developed countries, such as the Netherlands, Norway, Germany, England, and France. These studies have emphasized the usefulness of marine algae as functional cosmetic materials [24]. *Leathesia difformis* (L.) Areschoug (*L. difformis*) is a yellowish-brown alga that is found attached to rocks or other plants. Fucoidan from *L. difformis* has apoptotic [25], antiviral [26], and antioxidant (DPPH radical scavenging and superoxide scavenging) properties [27]. However, the effects of *L. difformis* on melanin synthesis are still understudied.

The purpose of this study is to examine the effects of *L. difformis* extract on melanogenesis in vitro. For this, an 80% (*v/v*) ethanol (EtOH) extract of *L. difformis* (LDE) was prepared, and then fractioned between ethyl acetate (EtOAc) (LDE-EA) and water (LDE-A). The melanin content and cellular tyrosinase activity of α-MSH-stimulated B16F10 melanoma cells were evaluated. In addition, it was investigated the effects of *L. difformis* extract on molecular mechanisms involved in the expression of melanin biosynthesis-related genes and proteins in B16F10 melanoma cells.

## 2. Results

### 2.1. Cytotoxicity of L. difformis Extracts in B16F10 Cells

The cell viability was determined using the 3-(4,5-dimethylthiazol-2-yl)-2,5-diphenyltetra-zolium bromide (MTT) assay. To investigate whether the *L. difformis* extracts exerted a cytotoxic effect on melanoma cells, B16F10 cells were treated with various concentrations (1–150 µg/mL) of the extracts. For comparison of minimum cytotoxic concentration of *L. difformis* extracts, the IC_20_ values, which represents 20% inhibitory concentration of cell viability, was determined. From results, IC_20_ for LDE and LDE-EA was 59.12 µg/mL and 19.54 µg/mL, respectively, while the IC_20_ of LDE-A was > 239.8 µg/mL at the highest dose (Figure 1A–C). The results showed that LDE-A had no significant effect on the cell viability. However, the other fractions had a cytotoxic effect on the melanoma cells, although the effect was not significant, and LDE-EA was found to be more cytotoxic than LDE. As shown in Figure 1D,E, no cytotoxicity was observed for the B16F10 cells treated with concentrations of LDE of < 19.54 µg/mL for 72 h. As a positive control, arbutin had no effect on the cell viability. From these results, the LDE concentrations of 1, 5, 10, and 15 µg/mL were selected for further studies on the melanin content, cellular tyrosinase activity, and melanogenesis related-gene expression in the α-MSH-stimulated B16F10 cells.

### 2.2. Effects of L. difformis Extracts on the Melanin Synthesis

To assess the inhibitory effect of the *L. difformis* extracts on melanin synthesis, we determined the melanin content of the B16F10 cells 72 h after treatment with three fractions of the extract concentration (1, 5, 10, and 15 µg/mL). The inhibitory effect of the *L. difformis* extracts on the melanin content is shown in Figure 2, which is represented by images of B16F10 cell pellets lysed with 1 N NaOH (10% (*v/v*) DMSO). The melanin content of the α-MSH-stimulated B16F10 cells treated with LDE-EA significantly decreased in a concentration-dependent manner (Figure 2B). Compared with the control, the melanin content (%) was 323.2%, 96.2%, and 135.6% for the α-MSH-stimulated control, positive control, and maximum concentration of LDE-EA, respectively. In addition, the color of the cell pellets lysed with 1 N NaOH (10% (*v/v*) DMSO) was changed in a dose-dependent manner, as a result of the change in the melanin content (Figure 2B). However, LDE and LDE-A showed less inhibitory activity toward the melanin content than LDE-EA (Figure 2A,C).

### 2.3. Effects of the L. difformis Extracts on Tyrosinase Activity

The effects of *L. difformis* extracts on the cellular tyrosinase activity in the α-MSH-stimulated B16F10 melanoma cells are shown in Figure 3A–C. These results showed that LDE and LDE-A had no effect on the intracellular tyrosinase activity. However, the cellular tyrosinase activity was significantly inhibited by LDE-EA in a dose-dependent manner (Figure 3B). When compared to the control, the tyrosinase activity (%) was 177.4%, 80.3%, and 113.2% for the α-MSH-stimulated control, arbutin as the positive control, and the maximum concentration of LDE-EA, respectively. These results were consistent with those that compared the effects of LDE-EA on the melanin content of B16F10 cells.

### 2.4. Effects of LDE-EA on Melanogenesis-Related Gene Expression

To evaluate whether the *L. difformis* extracts affected the expression of melanogenesis-related genes, such as *Tyr*, *Trp-1, Dct, Mitf*, and *Mc1r,* gene expression was examined in α-MSH melanoma cells using real-time qPCR. As the above experiments had shown that only LDE-EA had a significant effect on melanin synthesis, gene expression was only evaluated with this fraction. As shown in Figure 4, the mRNA levels of *Trp-1, Dct, Mitf*, and *Mc1r* decreased significantly in a dose-dependent manner, following treatment with 1, 5, 10, and 15 μg/mL of LDE-EA. However, the tyrosinase mRNA expression was only reduced at the maximum concentration of 15 μg/mL. In general, α-MSH induces *Mitf* expression, and the *Mitf* downstream genes, such as *Tyr*, *Trp-1*, and *Dct,* increase melanogenesis. Therefore, these results suggested that LDE-EA down-regulates the expression of *Tyr*, *Trp-1, Dct, Mitf*, and *Mc1r* mRNA to affect melanogenesis in α-MSH-stimulated melanoma cells.

### 2.5. Effects of LDE-EA on the Melanogenesis-Related Signaling Pathway

It was observed that α-MSH increases the transcriptional activity of MITF through the cAMP/CREB signaling pathway and promotes melanogenesis through the phosphorylation of MITF. To examine the phosphorylation activity of CREB in the cAMP/CREB pathway, due to LDE-EA treatment, a western blotting analysis for the expression of the p-CREB protein was implemented for the B16F10 cells treated with LDE-EA. As shown in Figure 5A–C, the protein level of β-actin, used as an internal control, showed no change. The total amount of the CREB protein decreased slightly at 10 and 15 μg/mL. The phosphorylation of CREB was down-regulated after a high dose of treatment with LDE-EA. Also, because LDE-EA inhibited melanin synthesis and tyrosinase expression, we evaluated if LDE-EA is reduced the production of cAMP, which induces the expression of MITF, in B16 melanoma cells. As shown in Figure 5D, we found that cAMP levels were decreased by LDE-EA in a dose dependent manner. These results indicate that the inhibitory effect of LDE-EA on melanin synthesis can be mediated to MITF degradation via the cAMP/PKA/CREB pathway.

### 2.6. Analysis of Fucoxanthin in LDE-EA by HPLC

Fucoxanthin (3′-acetoxy-5,6-epoxy-3,5′-dihydroxy-6′,7′-didehyro-5,6,7,8,5′,6′-hexahydro-β,β-carotene-8-one) is a carotenoid with various bioactive properties found in brown algae. The chromatogram of LDE-EA fraction is shown in Figure 6. The observed peak was assigned as fucoxanthin by comparing their retention time with standard compound in LDE-EA extract in the chromatogram at 280 nm. There are 0.15 g fucoxanthin per 1 kg of the extract. As can be seen in Figure 6, there is also another major compound besides the peak of fucoxanthin, which could be another bioactive component of this extract.

## 3. Discussion

In recent years, cosmetics, health foods, and drugs, including those containing algae-derived ingredients, are preferred due to the various bioactive compounds, as well as the rich source of nutrients [28,29,30]. In particular, brown algae are rich in physiologically active ingredients, such as fucoidan, pyropheophytin, and phlorotannin, and are used in various fields around the world. For example, fucoidan is used for hyperpigmentation or is used as a functional cosmetic material. Likewise, although the components of brown algae have been used in various fields, since their antioxidant, anti-inflammatory, anti-cancer, and anti-whitening effects have been revealed [31,32,33,34], there has been no research on the whitening effect of *L. difformis,* which is a type of brown algae, and the regulation of the melanin synthesis pathway. Therefore, we investigated the potential anti-melanogenesis properties of the *L. difformis* extracts (LDE, LDE-EA, and LDE-A), and the whitening effects of the extract on the melanin synthesis pathway were examined.

In this study, several fractions of *L. difformis* were investigated for their inhibitory effect on the melanin synthesis pathway and compared with arbutin as a positive control. Melanin is produced by a series of enzymatic reactions that involve *Mitf, Trp-1,* and *Dct*, with tyrosinase being the most important component of this process. Tyrosinase, which catalyzes the early stage of melanin biosynthesis, is converted to DOPA and dopaquinone in order to synthesize eumelanin and pheomelanin [35,36]. To confirm the effect of the *L. difformis* extracts on cellular melanogenesis, cell viability assays were performed to determine the concentration of the *L. difformis* extracts that did not cause cytotoxicity. Cells were treated with 0–150 μg/mL of the *L. difformis* extracts, and cytotoxicity was observed for LDE-EA in a concentration-dependent manner (Figure 1A-C). In the case of LDE, cytotoxicity of 14.02% was observed at the highest concentration, while LDE-A showed no cytotoxicity. Based on these results, treatment with the *L. difformis* extract at 0–15 μg/mL for 72 h showed no cytotoxicity (Figure 1D–F). For the comparative analysis, the concentrations for subsequent assays were therefore set at 1, 5, 10, and 15 μg/mL.

Melanin production by the B16F10 cells, following treatment with 1, 5, 10, and 15 μg/mL of the *L. difformis* extract, was reduced in a concentration-dependent manner without cytotoxicity, with only LDE-EA (Figure 2). When the data were expressed as the percentage normalized to the non-treated cells, the cells treated with α-MSH, arbutin, and the LDE-EA showed 323.2%, 96.2%, and 135.6% melanin production, respectively. The inhibitory effect of the *L. difformis* extract on melanin production was similar to that of the positive control arbutin, and the inhibition of the intracellular tyrosinase activity, which is implicated in melanin production, was also confirmed (Figure 3). Similar to the melanin production assay, the inhibition activity of the *L. difformis* extract was confirmed only for LDE-EA. These results demonstrate that LDE-EA inhibits melanin synthesis and down-regulates tyrosinase activity in α-MSH induced B16F10 cells.

To elucidate the mechanisms involved in the inhibition of melanin biosynthesis by *L. difformis*, we examined possible LDE-EA influences on the mRNA expression of melanogenesis-related genes and CREB activation. Generally, melanin is synthesized through several intracellular signal transduction pathways. As the primary mechanism for melanin synthesis, the cAMP/PKA pathway induces the expression of proopiomelanocortin (POMC) in keratinocytes when the skin is exposed to UV light. Increased levels of α-MSH then bind to the *Mc1r* receptor that is present in the cell membrane of melanocytes; this results in an increase in cAMP and the activation of PKA, a downstream signaling molecule, which increases *Mitf* expression through CREB. *Mitf* promotes the expression of tyrosinase, *Trp-1*, and *Dct* as important transcription factors for melanin synthesis [4,37,38,39,40,41]. Therefore, it is important to confirm the inhibition of LED-EA on melanogenesis by examining the down-regulation of MITF, tyrosinase, Trp-1, and Dct expression. Our results indicate that LDE-EA effectively down-regulated the mRNA expression of *Tyr*, *Trp-1, Dct, Mitf*, and *Mc1r*, suggesting that LED-EA does not directly inhibit tyrosinase activity. This effect can be explained by down-regulation of *Tyr*, *Trp-1*, and *Dct* through the inhibition of *Mitf* expression in B16F10 cells.

Furthermore, to evaluate the particular signaling pathway for melanogenesis, total CREB and the phosphorylation status of CREB were assessed by Western blot. Our data showed that the p-CREB level was decreased in the LDE-EA-treated melanoma cells. Also, we measured the intracellular cAMP levels in B16F10 cells stimulated by α-MSH. LDE-EA led to significant inhibition of cAMP levels in B16F10 cells stimulated by α-MSH (Figure 5D). These results indicate that LDE-EA can induce anti-melanogenesis through a cAMP dependent CREB-mediated pathway in B16F10 cells stimulated by α-MSH.

Transcriptional regulation of tyrosinase is mainly dependent on level of MITF, and MITF is up-regulated by CREB. Phosphorylated CREB up-regulates MITF, which binds to M-box and E-box motifs in the promoter of target genes related to melanin production [38]. Therefore, to examine whether LED-EA activates/inhibits CREB pathways, we assessed total CREB and the phosphorylation status of CREB by Western blot. Our data showed that the LDE-EA significantly inhibited activation of CREB. These results suggest that CREB signaling can play an important role in anti-melanogenesis of LED-EA treated B16F10 cells.

Previously, many types of brown algae-derived phytochemicals with hypopigmenting effects, including phlorotannins, carotenoids, meroterpenoids, and sulfated polysaccharides, have been reported [42]. Phlorotannins, derivatives of phloroglucinol (1,3,5-trihydroxybenzene), are polyphenolic compounds found only in brown algae [43]. Among these phlorothannins, eckol and dieckol from *Ecklonia cava* and 4-hydroxyphenethyl alcohol from *Hizikia fusiformis* were isolated from EtOAc fraction of MeOH or EtOH extract, respectively [44,45]. However, meroterpenoid compounds are mainly isolated from EtOH extract of the *Sargassum* genus [46,47] and fucoidan, a fucose-rich sulfated polysaccharide, is included in the crude extract of *Fucus vesiculosus* [48]. Fucoxanthin, which is one of carotenoids, was identified in organic solvent extracts, such as acetone [49].

Among these phytochemicals, fucoxanthin is a yellowish-brown pigment, which constitutes ~70% of the carotenoid found in brown algae [50]. Fucoxanthin is reported to have anti-oxidative, anti-inflammatory, anti-cancer, and anti-obese properties [51,52,53,54]. Also, anti-melanogenic activities of fucoxanthin were documented by Simoda et al. [55]. Therefore, we determined whether fucoxanthin is one of the constituents of LDE-EA through HPLC analysis. HPLC analysis of LDE-EA showed that LDE-EA includes fucoxanthin (Figure 6). Although our results suggest that the anti-melanogenic effect of LDE-EA on B16F10 cells may be attributed to fucoxanthin, other compounds present in LDE-EA may also have an effect on the overall ability to inhibit melanogenesis. In the present study, LED-EA was isolated from EtOAc fraction of the *L. difformis* EtOH extract, and it is similar with protocols used for isolation of phlorotannins in above mentioned literature studies [44,45]. Through further study, it is expected to be able to identify more complex derivatives of phlorotannins, such as eckol, with hypopigmenting effects. Even though this analyzed data is preliminary, it is crucial because there has been no report about *L. difformis* EtOAc extract regarding effect on anti-melanogenesis.

In addition, understanding the molecular mechanisms of a bioactive compound in specific targets is very important for its proper application. However, there are limited reports on detailed molecular events and specific targets for the anti-melanogenic effects of these brown algae-derived compounds. As reported in previous literature studies, fucoxanthin from *Laminaria Japonica* suppresses *Pge2*, *Msh*, and *Trp-1*, and melanogenic stimulant receptors, p75 neurotrophin receptor (NTR), prostaglandin EP1 receptor (EP1), and MC1R in UV-irradiated mice [55]. Dioxinodehydroeckol, a phlorotannin isolated from *Ecklonia stolonifera*, is involved in hypopigmentation by phosphatidylinositol 3-kinase (PI3K)/AKT-mediated down-regulation of *Mitf* [56]. It is reported that octaphlorethol A from *Ishige foliacea* inhibits melanogenesis by the extracellular signal-regulated kinase (ERK) 1/2-mediated down-regulation of *Mitf*, *Tyr*, *Trp-1*, and *Trp-2* in B16F10 cells [57]. Fucoidan, a fucose-rich sulfated polysaccharide isolated from *Fucus vesiculosus*, is involved in hypopigmentation by ERK-mediated down-regulation of *Mitf* in Mel-Ab cells [58]. Azam et al. elucidated that ethanolic extract from *Sargassum serratifolium* included three active meroterpenoid compounds, namely sargahydroquinoic acid, sargaquinoic acid, and sargachromenol, and this extract inhibited hyperpigmentation in B16F10 cells through regulation of *Mitf* via cAMP/CREB and ERK signaling pathways [47]. In this study, it is elucidated that LDE-EA from *L. difformis* can inhibit the melanogenesis by the decreased gene expression of *Tyr*, *Trp-1*, and *Dct* through the inhibition of *Mitf* expression via cAMP/CREB signaling pathway regulation (Figure 7).

This study is worthy of being the first study to identify the mechanism on anti-melanogenic effect of *L. difformis*, a brown alga is yet not elucidated the mechanism of action. Moreover, it is important to present the possibility of the identification of a new potential compound with hypopigmenting activity. Therefore, LDE-EA may be considered a potential therapeutic agent for the treatment of skin-pigmentation related diseases and may be a useful component in the skin-whitening cosmetic industry.

## 4. Materials and Methods

### 4.1. Chemicals and Reagents

Dimethyl sulfoxide (DMSO), α-MSH, NaOH, MTT, arbutin, synthetic melanin, Triton X-100, radioimmunoprecipitation assay (RIPA) buffer, skim milk, Tween 20, L-DOPA, and fucoxanthin analytical standard were obtained from Sigma-Aldrich (St. Louis, MO, USA). Dulbecco’s modified Eagle’s medium (DMEM), fetal bovine serum (FBS), penicillin/streptomycin, trypsin-ethylenediaminetetraacetic acid, TRIzol solution, and bicinchoninic acid (BCA) protein assay kit were purchased from Thermo Fisher Scientific (Waltham, MA, USA). Antibodies against CREB, p-CREB, β-actin, and anti-rabbit horseradish peroxidase antibody were purchased from Cell Signaling Technology (Danvers, MA, USA). Primers against *Tyr, Trp-1, Dct, Mitf, Mc1r,* and *GAPDH* genes were synthesized by Bioneer (Daejeon, Korea). An enhanced chemiluminescence (ECL) kit and polyvinylidene fluoride (PVDF) membrane were obtained from Bio-Rad (Hercules, CA, USA). EtOH and EtOAc from DaeJung Chemicals and Metals Co., Ltd. (Siheung, Korea) were analytical grade.

### 4.2. Preparation of L. difformis Extract

The *L. difformis* powder was purchased from PARAJEJU (Jeju, Korea). *L. difformis* extract was prepared with modification of extraction protocol by Kang et al. [44]. For the EtOH extract, the powdered *L. difformis* (1 kg dried weight) was first soaked with 80% (*v/v*) EtOH (10 L) in deionized water (D.W.) at room temperature for 24 h. Then, it was filtered by Whatman filter paper and the solvents of filtrates were evaporated by the vacuum rotary evaporator. After freeze-drying, the 80% EtOH extract from *L. difformis* was lyophilized to yield 232 g of extract. The extract was partitioned between to EtOAc-water (1:1 ratio, *w/v*) to give EtOAc-soluble fraction and water soluble fraction. The EtOAc fraction was concentrated in a rotary evaporator. The EtOAc and water fractions were dried using vacuum evaporator to obtain 10.61 g of EtOAc and 178.2 g of H_2_O, respectively. The *L. difformis* extracts were dissolved in DMSO.

### 4.3. Cell Culture

The mouse melanoma cell line B16F10 (CRL-6475) was purchased from American Type Culture Collection (ATCC, Rockville, MD, USA) and maintained in a humidified atmosphere of 5% CO_2_ and 95% air at 37 °C. The culture medium used was DMEM supplemented with 10% (*v/v*) heat-inactivated fetal bovine serum plus 100 U/mL-100 µg/mL penicillin-streptomycin. The medium was renewed every two days.

### 4.4. Cell Viability Assay

Cell viability was determined by the MTT assay. B16F10 cells were plated into a 96-well plate at a density of 5 × 10^4^ cells/mL for 24 h and treated with various concentrations from 1 to 150 µg/mL of *L. difformis* extract for 24 h at 37 °C in 95% humidified air and 5% CO_2_. To compare cells with melanin contents assay, the cells were seeded in a 24-well plate at a concentration of 1 × 10^4^ cells/mL for 72 h. Following the incubation, the cells were treated with MTT (0.5 mg/mL in phosphate-buffer saline, PBS) for 3 h. The medium was then removed and 1 mL of DMSO was added into each well to dissolve formazan crystals, the metabolite of MTT. After thoroughly mixing, the absorbance was measured at 570 nm using a microplate reader that is directly correlated with cell quantity. The percentage of cells showing cytotoxicity was determined relative to the control group.

### 4.5. Measurement of Melanin Contents

Melanin contents was executed as described, with minor modifications [59]. The B16F10 melanoma cells were seeded at a density of 2 × 10^4^ cells per mL in 24-well culture plates and then incubated for 24 h. The cells were exposed to various concentrations (1, 5, 10, and 15 µg/mL) of the *L. difformis* extracts or 1 mM arbutin for 72 h in the presence or absence of 200 nM α-MSH. The cells were washed twice in phosphate-buffered saline and lysis dissolved in 1 N NaOH (in 10% (*v/v*) DMSO in D.W.) by 1 h at 80 °C. The absorbance at 405 nm was measured using a microplate reader. The melanin contents were determined from a standard curve prepared from an authentic standard of synthetic melanin. In addition, the melanin contents were determined based on the absorbance/µg of protein in the extract from each cell. The protein concentration was determined by the BCA protein assay kit using BCA as a standard.

### 4.6. Cellular Tyrosinase Activity

Cellular tyrosinase activity was determined as described previously, with slight modifications [60]. The culture method for determining cellular tyrosinase assay was similar to that for determining melanin contents. The B16F10 melanoma cells were seeded at a density of 2 × 10^4^ cells/mL in 24-well culture plates and then incubated for 24 h. Cells were then exposed to increasing doses of *L. difformis* extracts or arbutin for 72 h in the presence or absence of 200 nM α-MSH. The cells were then washed with phosphate buffered saline (pH 6.8) and lysed with 1 % (*v/v*) Triton X-100. The pellet solutions were frozen and thawed three times and then centrifuged at 13,000 rpm for 15 min at 4 °C. The protein concentration was determined by the BCA protein assay kit using BCA as a standard. The reaction mixture consisting of 40 μg protein (adjusted to 100 µL with 1% Triton X-100) and 100 µL of 5 mM L-DOPA was added to each well of a 96-well plate. After incubation at 37 °C for 1 h, the optical density at 475 nm (OD_475_) was measured using a microplate reader. Tyrosinase activity in the protein was calculated by the following formula:
$$\mathrm{Tyrosinase}\;\mathrm{activity}\;(\%)=\left(\frac{{\mathrm{OD}}_{475\;}\mathrm{of}\;\mathrm{with}\;\mathrm{extract}\;\mathrm{treatment}}{{\mathrm{OD}}_{475}\;\mathrm{of}\;\mathrm{without}\;\mathrm{extract}\;\mathrm{treatment}}\right)\times 100$$

### 4.7. RNA Isolation and Real—Time PCR

The expressions of *Tyr*, *Trp-1, Dct, Mitf*, and *Mc1r* genes were determined by real-time PCR using glyceraldehyde 3-phosphate dehydrogenase (*Gapdh*) as an internal positive control. Melanoma cells (1 × 10^5^ cells/mL) were plated on 6-well plates and incubated. Then, the cells were treated with various concentrations of *L. difformis* extracts for 24 h in the presence or absence of 200 nM of α-MSH. The cells were harvested and washed twice with PBS. Total cellular RNA was prepared using TRIzol solution according to the manufacturer’s instructions. RNA was converted to cDNA using ReverTra Ace^®^ qPCR RT master mix with gDNA remover (TOYOBO, Osaka, Japan), according to the manufacturer’s instructions. The THUNDERBIRD SYBR^®^ qPCR Mix (TOYOBO, Osaka, Japan) was used in all the samples and reactions were carried out in a 15 μL final reaction volume. Each experiment was performed at least twice in duplicates using 10 μg of the following primers in Table 1.

### 4.8. Western Blot Analysis

B16F10 cells (1 × 10^5^ cells/mL) were treated with different concentration of *L. difformis* on 6-well plates. After treatment, the cells were collected and lysed using RIPA buffer. The lysates as protein sample were denatured at 95 °C. Protein concentrations of cell lysates were determined by the protein assay kit. Equal amounts of protein were separated using 10% (*w/v*) SDS-polyacrylamide gel electrophoresis (SDS-PAGE) running gel and transferred to a polyvinylidene fluoride (PVDF) membrane. The membrane was blocked using 5% (*w/v*) skim milk with Tris buffered saline containing 0.05% (*v/v*) Tween 20 (TBS-T). Membranes were incubated with different primary antibodies for overnight including CREB, p-CREB (1:2000), and β-actin (1:3000), and further incubated with anti-rabbit horseradish peroxidase antibody for 1 h. The bands of bound antibodies were detected by enhanced chemiluminescence reagents. Loading control was assessed using anti-β-actin antibody. All determinations were performed in triplicate.

### 4.9. cAMP Measurement Assay

The cAMP levels were measured using a cAMP ELISA kit (Cayman Chemical, Ann Arbor, MI, USA). Briefly, B16F10 cells (3 × 10^5^) were lysed in 0.1 M HCl to inhibit the phosphodiesterase activity. The supernatants were then collected, neutralized, and diluted. After neutralization and dilution, a fixed amount of cAMP conjugate was added to compete with cAMP in the cell lysate for sites on rabbit polyclonal antibody immobilized on a 96-well plate. After washing to remove excess conjugated and unbound cAMP, substrate solution was added to the wells to determine the activity of the bound enzyme. The color development was then stopped, after which the absorbance was read at 415 nm. The intensity of the color was inversely proportional to the level of cAMP in the cell lysate.

### 4.10. HPLC Analysis

The HPLC analysis was carried out on a Waters system (Waters Corp., Milford, MA, USA), consisting of a separation module (e2695) with an integrated column heater, an autosampler, and a photodiode array detector (2998). UV absorbance was monitored at 200 to 400 nm. Quantification was carried out by integration of the peak areas at 280 nm. The sample and standard injection volume was 10 μL. A C18 column (250 × 4.6 mm; particle size, 5 um; YMC Co. Ltd., Kyoto, Japan) was installed in a column oven and maintained at 30 °C. The mobile phase was composed of water containing 0.5% (*v*/*v*) tri-fluoro acetic acid (solvent A) and acetonitrile (solvent B). The flow rate was 1.0 mL/min. The gradient was 20 % A and 80 % B. LDE-EA and fucoxanthin were used as sample and standard at a concentration of 10 mg/mL and 10 μ/mL of each materials for HPLC analysis to identify the peaks.

### 4.11. Statistical Analysis

The data were analyzed by using Statistical Analysis System (SAS) software (PRISM) (GraphPad Software, San Diego, CA, USA). All data are expressed as mean ± S.E.M. Statistical comparisons between different treatments were performed by using one-way ANOVAs with Tukey’s multiple comparison post-hoc tests. The * *p*-values < 0.05 indicated a statistical significance.

## 5. Conclusions

This study demonstrated, for the first time, the inhibitory effects of *L. difformis* extracts on melanin biosynthesis. The major finding is that LDE-EA can inhibit the melanogenesis by the decreased gene expression of *Tyr*, *Trp-1*, and *Dct* through the inhibition of *Mitf* expression. In addition, LDE-EA exerts an inhibitory effect on melanogenesis through cAMP/CREB signaling pathway regulation (Figure 7). Therefore, LDE-EA may be considered a potential therapeutic agent for the treatment of skin-pigmentation related diseases and may be a useful component in the skin-whitening cosmetic industry.

## Figures and Tables

**Figure 1 ijms-20-00536-f001:**
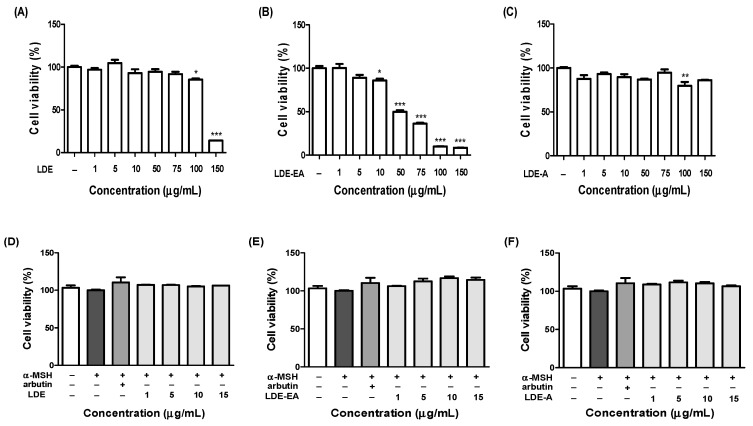
Effect of *L. difformis* extract on B16F10 melanoma cell viability. Cells were treated with various concentrations of LDE (**A**), LDE-EA (**B**), and LDE-A (**C**) for 24 h, and were then treated with 200 nM α-MSH and 1, 5, 10, and 15 µg/mL of LDE (**D**), LDE-EA (**E**), and LDE-A (**F**) for 72 h. Arbutin was used as a positive control at a concentration of 1 mM. Cell viability was measured by MTT assay. The results are represented as a percentage of control. Values are represented as the mean ± SEM of three independent experiments; * *p* < 0.05, ** *p* < 0.01, and *** *p* < 0.001 versus control.

**Figure 2 ijms-20-00536-f002:**
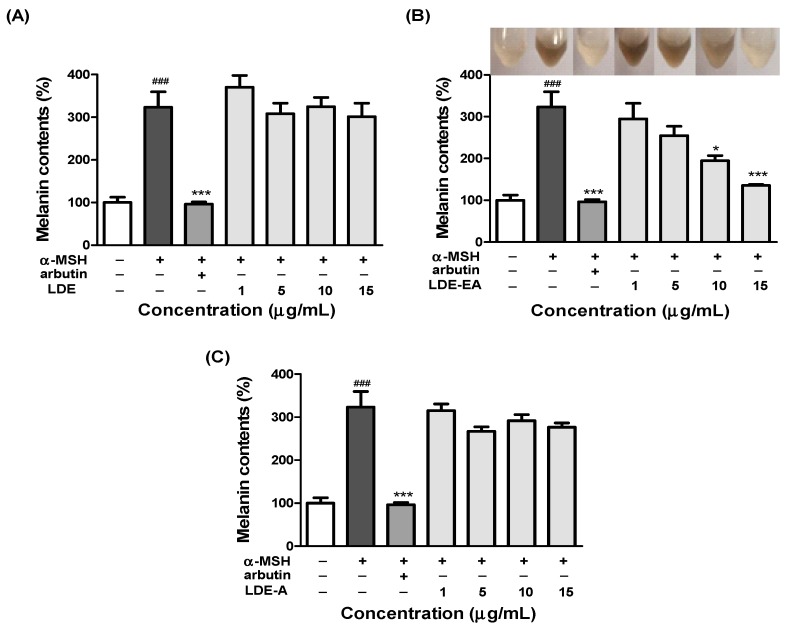
Effect of *L. difformis* extract on melanogenesis in B16F10 cells. B16F10 cells were exposed to 200 nM α-MSH in the presence of 1, 5, 10, and 15 μg/mL *L. difformis* extract ((**A**) LDE, (**B**) LDE-EA, and (**C**) LDE-A) or 1 mM arbutin (a melanin inhibitor). Each percentage value for the treated cells was reported relative to that of the control cells. Values are represented as the mean ± SEM of three independent experiments. Note: ### *p*<0.001 compared with the control; * *p* < 0.05, and *** *p* < 0.001 compared with the α-MSH-treated control.

**Figure 3 ijms-20-00536-f003:**
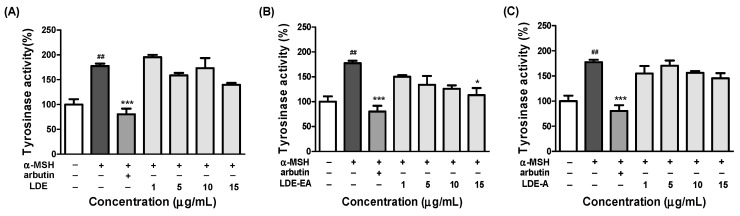
Effect of *L. difformis* extract on cellular tyrosinase activity in B16F10 cells. Cells were exposed to 200 nM α-MSH in the presence of 1, 5, 10, and 15 μg/mL *L. difformis* extract ((**A**) LDE, (**B**) LDE-EA, and (**C**) LDE-A) or 1 mM arbutin. Each percentage value for the treated cells was reported relative to that of the control cells. Values are represented as the mean ± SEM of three independent experiments. Note: ## *p* < 0.01 compared with the control; * *p* < 0.05, and *** *p* < 0.001 compared with the α-MSH-treated control.

**Figure 4 ijms-20-00536-f004:**
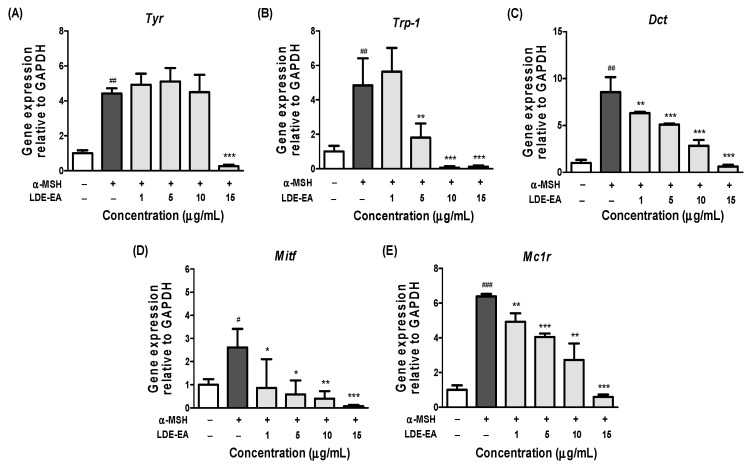
Effects of LDE-EA on mRNA expression of *Tyr*, *Trp-1, Dct, Mitf*, and *Mc1r* in B16F10 cells stimulated with α-MSH. B16F10 cells were co-treated with LDE-EA and 200 nM α-MSH. After treatment, the mRNA expression of *Tyr* (**A**), *Trp-1* (**B**), *Dct* (**C**), *Mitf* (**D**), and *Mc1r* (**E**) was measured and normalized to GAPDH expression. Values are represented as the mean ± SEM of three independent experiments. Note: # *p* < 0.05, ## *p* < 0.01, and ### *p* < 0.001 compared with the control group; * *p* < 0.05, ** *p* < 0.01, and *** *p* < 0.001 compared with the α-MSH-treated group.

**Figure 5 ijms-20-00536-f005:**
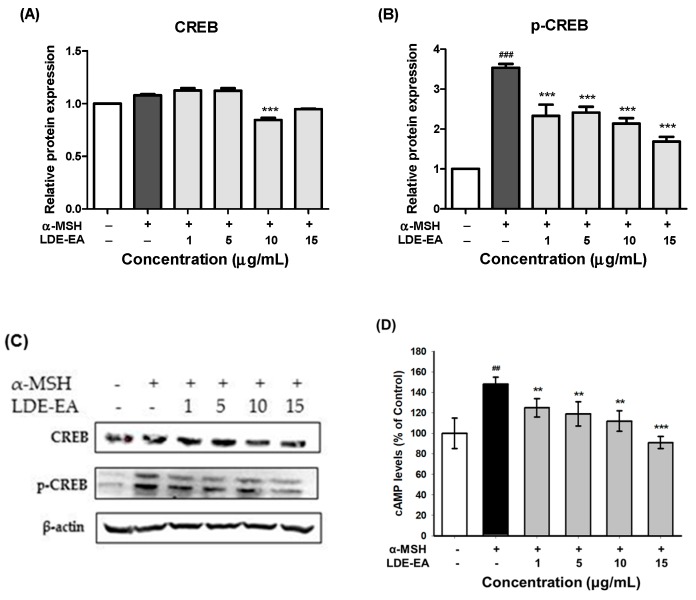
Effects of LDE-EA on protein expression of p-CREB and cAMP levels in B16F10 cells. Cells (1 × 10^5^ cells/mL) were cultured for 24 h, and treated with LDE-EA and 200 nM α-MSH for 3 days. After treatment, total cell lysates were assayed by western blot analysis using antibodies for CREB (**A**,**C**) and p-CREB (**B**,**C**). Equal amounts of protein loading were checked using β-actin antibodies. (**D**) The total amount of cellular cAMP was assayed using the immunoassay kit. Values are represented as the mean ± SEM of three independent experiments. Note: ### *p* < 0.001 and ## *p* < 0.01 compared with the control group; *** *p* < 0.001 and ** *p* < 0.01 compared with the α-MSH-treated group.

**Figure 6 ijms-20-00536-f006:**
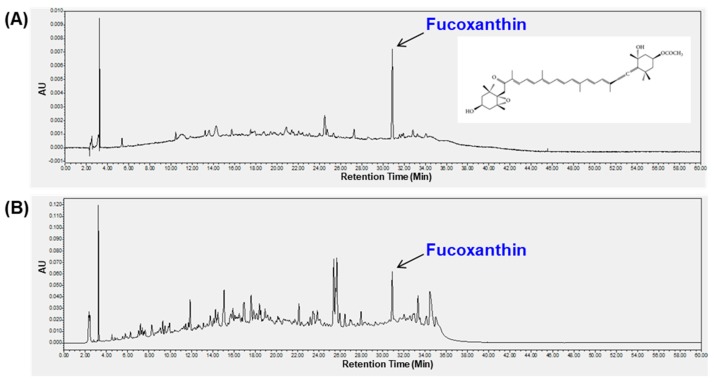
High-performance liquid chromatography (HPLC) chromatograms of (**A**) the fucoxanthin standard (**B**) EtOAc fraction of *L. difformis* EtOH extract (LDE-EA). AU indicates the absorbance unit.

**Figure 7 ijms-20-00536-f007:**
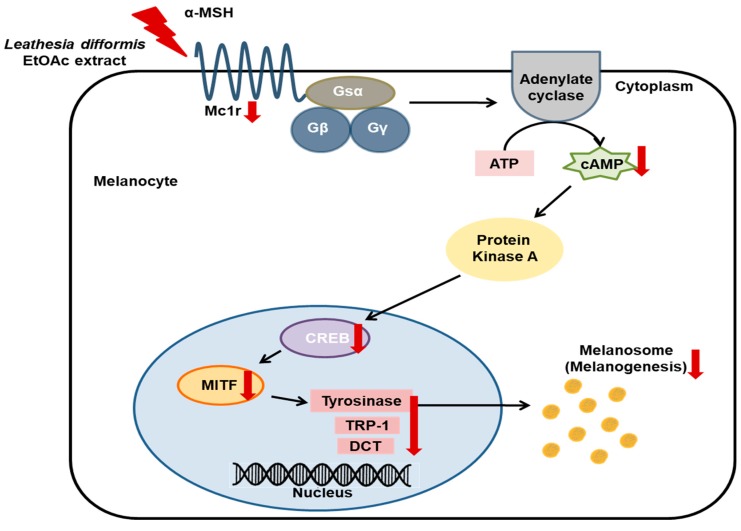
Proposed mechanism of EtOAc extract of *L. difformis* (LDE-EA) for inhibition of α-MSH-induced melanogenesis. Red polyline arrow indicates the treatment to melanocyte. Red arrows indicate the activity of LDE-EA and black arrows indicate the direction of regulation.

**Table 1 ijms-20-00536-t001:** Sequences of the primer pairs of specific target genes.

Target Gene	Sequence	
*Tyr*	Forward	AAGAATGCTGCCCACCATGG
	Reverse	CACGGTCATCCACCCCTTTG
*Trp-1*	Forward	CAGTGCAGCGTCTTCCTGAG
	Reverse	TTCCCGTGGGAGCACTGTAA
*Dct*	Forward	GATGGCGTGCTGAACAAGGA
	Reverse	ATAAGGGCCACTCCAGGGTC
*Mitf*	Forward	ATCCCATCCACCGGTCTCTG
	Reverse	CCGTCCGTGAGATCCAGAGT
*Mc1r*	Forward	TCATCGTCCTCTGCCCTCAG
	Reverse	GCAGCACCTCCTTGAGTGTC
*Gapdh*	Forward	TTGGCATTGTGGAAGGGCTC
	Reverse	ACCAGTGGATGCAGGGATGA

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
