# Peer review of "Leathesia difformis Extract Inhibits α-MSH-Induced Melanogenesis in B16F10 Cells via Down-Regulation of CREB Signaling Pathway"

_ijms, 2019, doi:10.3390/ijms20030536_

Reviewer 1 Report

The reviewed manuscript is in accordance with the aim and scope of the Journal. The subject is interesting and worthy of investigation, the manuscript is well organized and described. However, it seems that the studies are preliminary and their greatest weakness is lack of characterization of the chemical composition of the studied fractions. Plant material is very variable, therefore, even for preliminary study the chemical composition is crucial.

Moreover, I have some suggestions and questions to explain before its publication. The specific list is below:

1)      Some information are missing, e.g.

-          The type of % concentration should be given e.g. m/m, v/m or ... .  It should be added in all cases.

-          Was ethanol and DMSO mixed in water (80% ethanol in..., 10% DMSO in....)? It should be clarified.Line 90 page 2, why did the Authors use IC20? It should be explained.

-          Was the extraction procedure base on literature? This should be clarified.

-          Line 249 page 8, the time of ultrasonication should be given.

-          Line 252 page 8, was the water fraction dried in a rotary evaporator? It is not clear

-          Some important information in Experimental section are missing. All reagents used for the experiment, their purity and manufacturer, should be added e.g. EtOH, NaOH,…..

-          The abbreviation should be explained first e.g. line 298 page 9 (OD475). Is the value 475 correct?

-          There are some editorial errors e.g. in title: lack of italic, redundant capital letter, ….

Author Response

Response to Reviewer 1 Comments

Our response to the reviewers’ comments and the corresponding text from the revised manuscript are presented below in red and blue fonts, respectively.

Major Comment:

Point 1: The reviewed manuscript is in accordance with the aim and scope of the Journal. The subject is interesting and worthy of investigation, the manuscript is well organized and described.

Response 1: We would like to thank the reviewer for careful and thorough reading of this manuscript and for the thoughtful comments and constructive suggestions, which help to improve the quality of this manuscript.

Point 2: However, it seems that the studies are preliminary and their greatest weakness is lack of characterization of the chemical composition of the studied fractions. Plant material is very variable, therefore, even for preliminary study the chemical composition is crucial.

Response 2: The reviewer brings up an important point. We agree that the characterization of chemical of LDE-EA fraction is important and this point is not presented in old version of manuscript. We have been afforded to analyse the compounds of LDE-EA fraction using various techniques. However, the characterization of chemical is difficult due to various constraints, such as the absence of standard materials for analytical purpose and the absence of database of MS spectrum from brown algae.  Despite these difficult conditions, we analysed on one candidate compound, fucoxanthin, in LDE-EA fraction using HPLC. According to the reviewer’s comment that it is crucial even though analysed result of chemical composition is preliminary, we added and discussed the result of HPLC analysis of LDE-EA fraction. Of course, although we could find fucoxanthin in LDE-EA in this analysis, there is also another major compound besides the peak of fucoxanthin which can be another bioactive component of this extract. Through this preliminary study, we can recognize the necessity of the characterization of chemical composition of our fraction from brown algae, so that further analysis is performing to identify chemical structure and composition of unknown compounds by comparison of HPLC spectrum of three extract fractions with different anti-melanogenic activities.

We added HPLC result in “2.6. Analysis of fucoxanthin in LDE-EA by HPLC” and “Figure 6” in result part, Discussion part, and Method part in the revised manuscript, as follows:

- Page 6 Line 187;

   2.6. Analysis of fucoxanthin in LDE-EA by HPLC

Fucoxanthin (3′-acetoxy-5,6-epoxy-3,5′-dihydroxy-6′,7′-didehyro-5,6,7,8,5′,6′-hexahydro-β,β-carotene-8-one) is a carotenoid with various bioactive properties found in brown algae. The chromatogram of LDE-EA fraction is shown in Figure 6. The observed peak was assigned as fucoxanthin by comparing their retention time with standard compound in LDE-EA extract in the chromatogram at 280 nm. There are 0.15 g fucoxanthin per 1 kg of the extract. As it can be seen in Figure 6, there is also another major compound besides the peak of fucoxanthin which can be another bioactive component of this extract.

- Page 7 Line 195;                                   

Figure 6. High-performance liquid chromatography (HPLC) chromatograms of (A) the fucoxanthin standard (B) EtOAc fraction of L. difformis EtOH extract (LDE-EA).

And also, we discussed on brown algae-derived phytochemicals with hypopigmenting effects reported in previous studies in Discussion part in the revised manuscript, as follows:

- Page 8 Line 261;

Previously, many types of brown algae-derived phytochemicals with hypopigmenting effects, including phlorotannins, carotenoids, meroterpenoids, and sulfated polysaccharides, have been reported [43]. Phlorotannins, derivatives of phloroglucinol (1,3,5-trihydroxybenzene) are polyphenolic compounds found only in brown algae [44]. Among these phlorothannins, eckol and dieckol from Ecklonia cava and 4-hydroxyphenethyl alcohol from Hizikia fusiformis were isolated from EtOAc fraction of MeOH or EtOH extract, respectively [45, 46]. However, meroterpenoid compounds is mainly isolated from EtOH extract of the Sargassum genus [47, 48] and fucoidans, a fucose-rish sulfated polysaccharide, is included in the crude extract of Fucus vesiculosus [49]. And fucoxanthin, which is one of carotenoids, was identified in organic solvent extracts, such as acetone [50].

Among these, fucoxanthin is a yellowish-brown pigment, which constitutes ~70% of the carotenoid found in brown algae [51]. Fucoxanthin is reported that it has anti-oxidative, anti-inflammatory, anti-cancer, and anti-obese properties [52, 53, 54, 55]. And also, anti-melanogenic activities of fucoxanthin were documented by Simoda et al. [56]. Therefore, we determined whether fucoxanthin is one of the constituents of LDE-EA through HPLC analysis. HPLC analysis of LDE-EA showed that 1 kg of the extract contained about 1.1 g of fucoxanthin (Figure 6). Although our results suggest that the anti-melanogenic effect of LDE-EA on B16F10 cells may be attributed to fucoxanthin, other compounds present in LDE-EA may also have an effect on the overall ability to inhibit melanogenesis. In the present study, LED-EA was isolated from EtOAc fraction of the L. difformis EtOH extract, and it is similar with protocols used for isolation of phlorotannins in above mentioned literatures [45, 46]. Through further study, it is expected to be able to identify more complex derivatives of phlorotannins, such as eckol, with hypopigmenting effects. Even though this analyzed data is preliminary, it is crucial because there has been no report about L. difformis EtOAc extract regarding on anti-melanogenesis.

In addition, the goal of this paper is firstly to present the anti-melanogenic activity of L. difformis and also to elucidate the mechanism of action regarding on this effect. Especially,

to clarify the lack of research on the mechanistic aspects of brown algae-derived chemicals and to emphasize the importance of this paper which suggests mechanism on anti-melanogenesis of L. difformis extract, we discussed on action mechanisms of brown algae-derived phytochemicals with hypopigmenting effects in 3. Discussion part in the revised manuscript, as follows:

- Page 9 Line 284;

In addition, understanding the molecular mechanisms of a bioactive compound in specific targets is very important for its proper application. However, there are limited reports on detailed molecular events and specific targets for the anti-melanogenic effects of these brown algae-derived compounds. As reported in previous literatures, fucoxanthin from Laminaria Japonica suppresses Pge2, Msh and Trp-1 and melanogenic stimulant receptors, NTR, EP1, and MC1R in UV-irradiated mice [56]. Dioxinodehydroeckol, a phlorotannin isolated from Ecklonia stolonifera is involved in hypopigmentation by phosphatidylinositol 3-kinase (PI3K)/AKT-mediated down-regulation of Mitf [57]. And it is reported that octaphlorethol A from Ishige foliacea inhibits melanogenesis by the extracellular signal-regulated kinase (ERK) 1/2-mediated down-regulation of Mitf, Tyr, Trp-1 & Trp-2 in B16F10 cells [58]. Fucoidan, a fucose-rich sulfated polysaccharide isolated from Fucus vesiculosus, is involved in hypopigmentation by ERK-mediated down-regulation of Mitf in Mel-Ab cells [59]. Azam et al., elucidated that ethanolic extract from Sargassum serratifolium was included of three active meroterpenoid compounds, including sargahydroquinoic acid, sargaquinoic acid and sargachromenol and this extract inhibited hyperpigmentation in B16F10 cells through regulation of Mitf via cAMP/CREB and ERK signaling pathways [48]. In this study, it is elucidated that LDE-EA from L. difformis can inhibit the melanogenesis by the decreased gene expression of Tyr, Trp-1, and Dct through the inhibition of Mitf expression via cAMP/CREB signaling pathway regulation (Figure 7).

This study is worthy of the first mechanistic study on anti-melanogenic effect of L. difformis, a brown alga is yet not elucidated the mechanism of action. Moreover, it is important to present possibility of the identification of a new potential compound with hypopigmenting activity. Therefore, LDE-EA may be considered a potential therapeutic agent for the treatment of skin-pigmentation related diseases and may be a useful component in skin-whitening cosmetic industry.

Minor Comments:

Moreover, I have some suggestions and questions to explain before its publication. The specific list is below:

Some information are missing, e.g.

Point 3: The type of % concentration should be given e.g. m/m, v/m or ... .  It should be added in all cases.

Response 3: We agree with the reviewer on this point. We added the type of % concentration in all cases of revised manuscript, as follows:

- Page 2 line 80;  

For this, an 80% (v/v) ethanol (EtOH) extract of L. difformis (LDE) was prepared,

- Page 3 line 114 and line 118;  

NaOH (10% (v/v) DMSO).

- Page 11 line 330;  

EtOH extract, the powdered L. difformis (1 kg dried weight) was first soaked with 80% (v/v) EtOH (10 L) in deionized water (D.W.)

- Page 11 line 344;  

supplemented with 10% (v/v) heat-inactivated fetal bovine serum

- Page 11 line 362;  

saline and lysis dissolved in 1 N NaOH (in 10% (v/v) DMSO in D.W.) by 1 h at 80°C.

- Page 13 line 400;  

using 10% (w/v) SDS-polyacrylamide gel electrophoresis (SDS-PAGE)

- Page 13 line 402;  

The membrane was blocked using 5% (w/v) skim milk with Tris buffered saline containing 0.05% (v/v) Tween 20 (TBS-T).

Point 4: Was ethanol and DMSO mixed in water (80% ethanol in..., 10% DMSO in....)?

Response 4:

Yes, we mixed EtOH and DMSO in deionized water. So, this point is clarified in the revised manuscript, as follows:

- Page 11 line 330;  

EtOH (10 L) in deionized water (D.W.) at room temperature for 24 h

- Page 11 line 362;  

saline and lysis dissolved in 1 N NaOH (in 10% (v/v) DMSO in D.W.) by 1 h at 80°C.

Point 5: It should be clarified. Line 90 page 2, why did the Authors use IC20? It should be explained.

Response 5: The reviewer brings up an important point. Generally, when we deal with cytotoxicity data obtained using cell-based assays (eg. MTT assay), IC20 (20% inhibitory concentration of cell viability) value means the minimum cytotoxic concentration, that just begins to cause measurable cellular injury, in case when dose-response relationship is found.  In this study, the reason we used the IC20 value is to determine the safe level of L. difformis extracts in the B16F10 cells and the actual used concentrations were also lower than the IC20 values of each fraction. This point is clarified in the revised manuscript, as follows:

- Page 2 line 90; 

For comparison of minimum cytotoxic concentration of L. difformis extracts, the IC20 values, which represents 20% inhibitory concentration of cell viability was determined.

Point 6: Was the extraction procedure base on literature? This should be clarified.

Response 6: We thank the reviewer for pointing out that we did not discuss this point. We were prepared the L. difformis extracts with modified protocol based on a previous literature (Kang et al. 2004) dealing with hypopigmenting compounds isolated from brown algae, E. stolonifera. This point is clarified in the revised manuscript, as follows:

- Page 11 line 327;

L. difformis extract was prepared with modification of extraction protocol by Kang et al. [45].  

Point 7: Line 249 page 8, the time of ultrasonication should be given.

Response 7: This point is obviously our mistake. Ultrasonication in process for preparation of extract was not performed. So, this point is removed in the revised manuscript, as follows:

- Page 11 line 329;

For the EtOH extract, the powdered L. difformis (1 kg dried weight) was first soaked with 80% (v/v) EtOH (10 L) in deionized water (D.W.) at room temperature for 24 h.

Point 8: Line 252 page 8, was the water fraction dried in a rotary evaporator? It is not clear.
Response 8: Both water fraction and EtOAc fraction were dried by vacuum evaporator. To clarify this point, we overall modified the preparation of L. difformis Extract in method part of the revised manuscript, as follows:

- Page 11 line 327;

4.2. Preparation of L. difformis Extract

The L. difformis powder was purchased from PARAJEJU (Jeju, Republic of Korea). L. difformis extract was prepared with modification of extraction protocol by Kang et al. [45]. For the EtOH extract, the powdered L. difformis (1 kg dried weight) was first soaked with 80% (v/v) EtOH (10 L) in deionized water (D.W.) at room temperature for 24 h. Then, it was filtered by Whatman filter paper and the solvents of filtrates were evaporated by the vacuum rotary evaporator. After freeze-drying, the 80% EtOH extract from L. difformis was lyophilized to yield 232 g of extract. The extract was partitioned between to EtOAc-water (1:1 ratio, w/v) to give EtOAc-soluble fraction and water soluble fraction. The EtOAc fraction was concentrated in a rotary evaporator. The EtOAc and water fractions were dried using vacuum evaporator to obtain 10.61 g of EtOAc and 178.2 g of H2O, respectively. The L. difformis extracts were dissolved in DMSO.

Point 9: Some important information in Experimental section are missing. All reagents used for the experiment, their purity and manufacturer, should be added e.g. EtOH, NaOH,…..

Response 9: We agree with the reviewer on this point. We added “Chemicals and Reagents” in the revised manuscript, as follows:

- Page 10 line 315;

4.1. Chemicals and Reagents

Dimethyl sulfoxide (DMSO), α-MSH, NaOH, MTT, arbutin, synthetic melanin, Triton X-100, radioimmunoprecipitation assay (RIPA) buffer, skim milk, Tween 20, L-DOPA, and fucoxanthin analytical standard were obtained from Sigma–Aldrich (St. Louis, MO, USA). Dulbecco’s modified Eagle’s medium (DMEM), fetal bovine serum (FBS), penicillin/streptomycin, trypsin–ethylenediaminetetraacetic acid, TRIzol solution, and bicinchoninic acid (BCA) protein assay kit were purchased from Thermo Fisher Scientific (Waltham, MA, USA). antibodies against CREB, p-CREB, β-actin, and anti-rabbit horseradish peroxidase antibody were purchased from Cell Signaling Technology (Danvers, MA, USA). Primers against Tyr, Trp-1, Dct, Mitf, Mc1r, and GAPDH genes were synthesized by Bioneer (Daejeon, Korea) An enhanced chemiluminescence (ECL) kit and polyvinylidene fluoride (PVDF) membrane were obtained from Bio-Rad (Hercules, CA, USA). EtOH and EtOAc from DaeJung Chemicals and metals co. ltd. (Siheung, Korea) were analytical grade.

Point 10: The abbreviation should be explained first e.g. line 298 page 9 (OD475). Is the value 475 correct?

Response 10: We agree with the reviewer on this point. We explained first all the abbreviation, including OD475 in the revised manuscript. And 490 nm written in the manuscript is incorrect, so we modified this point, as follows:

- Page 12 line 379;

After incubation at 37°C for 1 h, the optical density at 475 nm (OD475) was measured using a microplate reader. Tyrosinase activity in the protein was calculated by the following formula: 

Point 11: There are some editorial errors e.g. in title: lack of italic, redundant capital letter

Response 11: In the revised manuscript, some editorial errors were modified.

We hope to have addressed all reviewers’ suggestions to your complete satisfaction.

Sincerely,

Kim, Youn-Jung, Prof., PhD.

Reviewer 2 Report

The manuscript “Leathesia difformis extract inhibits α-MSH-induced 2 Melanogenesis in B16F10 cells via down-regulation of  CREB signaling pathway” documents that Leathesia difformis extracts have potential anti-melanogenic activity via the inhibition of the cAMP/PKA/CREB pathways. Before being acceptable for publication some minor medications are required.

Minor points:

1.       Proper nomenclature of Leathesia difformis (L.) Aresch needs to be followed

2.       Determine the levels of cAMP to show actual inhibition

3.       Discuss the possible inhibitory compounds in the Leathesia difformis extracts

4.       Figure 6 the melanogenesis pathway image in unreadable, either increase size or delete

Author Response

see the attachment

Round  2

Reviewer 1 Report

The manuscript has been carefully corrected by Authors. All disputable points have been explained, thus I recommend mentioned manuscript for publication